# Enhanced Air Stability of Perovskite Quantum Dots by Manganese Passivation

Ryota Sato [1], Kazuki Umemoto [1], Satoshi Asakura [2] and Akito Masuhara [1,3,*]

1. Graduate School of Science and Engineering, Yamagata University, 4-3-16, Yonezawa 992-8510, Yamagata, Japan; t211431d@st.yamagata-u.ac.jp (R.S.); umemoto@yz.yamagata-u.ac.jp (K.U.)
2. Ise Chemicals Corporation, 1-3-1, Kyobashi, Chuo-ku, Tokyo 104-0031, Japan; asakura@isechem.co.jp
3. Frontier Center for Organic Materials (FROM), Department of Yamagata University, 4-3-16, Yonezawa 992-8510, Yamagata, Japan
* Correspondence: masuhara@yz.yamagata-u.ac.jp; Tel.: +81-238-26-3891

**Abstract:** Organic-inorganic perovskite quantum dots (PeQDs) have attracted attention due to their excellent optical properties, e.g., high photoluminescence quantum yields (PLQYs; >70%), a narrow full width at half maximum (FWHM; 25 nm or less), and color tunability adjusted by the halide components in an entire tunability (from 450 nm to 730 nm). On the other hand, PeQD stability against air, humidity, and thermal conditions has still not been enough, which disturbs their application. To overcome these issues, with just a focus on the air stability, $Mn^{2+}$ ion passivated perovskite quantum dots (Mn/MAPbBr$_3$ QDs) were prepared. $Mn^{2+}$ could be expected to contract the passivating layer against the air condition because the $Mn^{2+}$ ion was changed to the oxidized Mn on PeQDs under the air conditions. In this research, Mn/MAPbBr$_3$ QDs were successfully prepared by ligand-assisted reprecipitation (LARP) methods. Surprisingly, Mn/MAPbBr$_3$ QD films showed more than double PLQY stability over 4 months compared with pure MAPbBr$_3$ ones against the air, which suggested that oxidized Mn worked as a passivating layer. Improving the PeQD stability is significantly critical for their application.

**Keywords:** perovskite quantum dot; manganese passivation; air stability

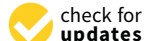



## 1. Introduction

Organic-inorganic hybrid lead halide perovskite ABX$_3$ (A = MA; methylammonium, FA; formamidinium, B = Pb, X = Cl, Br, I) quantum dots (PeQDs) have greatly attracted attention as a light source for optoelectrical applications carrying the next generation such as displays for remote medicine [1], photodetectors [2], and photovoltaics [3]. This is due to their excellent optical properties such as high photoluminescence quantum yields (PLQYs; over 70%), a narrow full width at half maximum (FWHM; 25 nm or less), and color tunability adjusted by the halide components in an entire tunability (from 450 nm to 730 nm) [4–7]. The major preparation method of PeQDs is hot injection [8,9] as follows: this approach is based on the rapid injection of a precursor into a hot solution of the precursors, ligands, and a high boiling solvent. The strongest feature of the prepared PeQDs by hot injection methods is a high mono-dispersity and size uniformity on the nanometer order due to the high temperature and N$_2$ atmosphere. However, this method has several demerits such as a high temperature of over 100 degrees Celsius and a lot of N$_2$ gas due to the high cost of fixating on the uniformity of the PeQD particle size, resulting in the disturbance of the development of the flow process. Therefore, in recent years, the ligand-assisted reprecipitation (LARP) method has attracted attention due to the simple method to prepare the PeQD colloidal solution through mixing the polar solvent (=perovskite precursor solution) and non-polar solvent at room temperature under air [10–12]. Similar to the hot injection method, PeQDs prepared by the LARP method could be achieved

with mono-dispersion and size uniformity, which means it is superior to the hot injection method in the point of its condition.

By using the LARP method, green-emissive PeQDs have especially improved their PLQYs by over 90% by filling their surface defects using alkyl ligands with certain chain lengths [13] and the surface passivation with metal halides [14]. In addition, by establishing the ligand exchanging and washing conditions of PeQDs to decrease the insulator components, the PeQD carrier mobility (=electron injection) was drastically improved, and green-emissive PeQDs with a champion external quantum efficiency (EQE) of 23.4% prepared by the LARP method were successfully obtained [15–17]. This means the prepared PeQDs have almost the same performance compared with the one prepared by the hot injection method [18]. On the other hand, the PeQD stability against air, humidity, and thermal conditions has not been enough [19], which has disturbed the application of PeQDs [20–22]. To solve these issues, establishing a strategy to enhance the stability of PeQDs against each condition is extremely significant.

Herein, our research group focused on the PeQD stability under the air condition. Here, "air" means under the conditions at 25 degrees Celsius and 25% relative humidity (RH). Our strategy is to construct a passivated layer on the surface of PeQDs using an $Mn^{2+}$ ion, which can possibly protect against the air [23]. The reason to choose an $Mn^{2+}$ ion is to be superior to a $Pb^{2+}$ ion in the point of the ion tendency. In detail, an $Mn^{2+}$ ion is more easily oxidized compared with a $Pb^{2+}$ ion due to the high ion tendency. With adding $MnBr_2$ when preparing PeQDs, an $Mn^{2+}$ ion was selectively attached to the PeQD surface, and it transformed from an $Mn^{2+}$ ion to oxidized Mn due to oxidation. Therefore, the oxidized Mn on PeQDs was expected to work as a blocking layer against the air, resulting in improving the PL stability of PeQDs under the air condition. This mechanism could not be achieved by the hot injection method because there is no oxygen to promote the oxidization of an $Mn^{2+}$ ion when preparing PeQDs. Therefore, the LARP method is the unique method to prepare PeQDs with a passivated layer using one-pot synthesis. From this background, we aimed to prepare PeQDs with high air stability by LARP methods for their application.

## 2. Materials and Methods

### 2.1. Materials

For preparing PeQDs, lead (II) bromide ($PbBr_2$, TCI from Tokyo, Japan), methylamine (40% in methanol, Wako from Osaka, Japan), manganese (II) bromide ($MnBr_2$, Wako, Osaka, Japan), hydrobromic acid (HBr, 48% *w/w*, Wako, Osaka, Japan), diethyl ether (99.5%, Wako, Osaka, Japan), ethanol (99.5%, Wako, Osaka, Japan), 1-methyl-2-pyrrolidone (NMP, ≥99.0% Wako, Osaka, Japan), ethyl acetate (99.5%, Wako, Osaka, Japan), toluene (99.5%, Wako), octylamine (OcAm, 99%, Sigma-Aldrich from St. Louis, MO, USA), and oleic acid (OlAc, ≥90%, Sigma-Aldrich, St. Louis, MO, USA) were used as received without further purification.

### 2.2. Synthesis of Methylammonium Bromide ($CH_3NH_2$ HBr; MABr)

MABr were synthesized as following the previous report with modification [9]. In detail, 7 mL of HBr solution was reacted with 30 mL of methylamine in a 250 mL round-bottom flask, and the solution was left under a slow stir for 2 h at 0 °C. Then, the solution was evaporated, and after that, white crystals were collected. These crystals were then washed with diethyl ether and recrystallized a third time in ethanol. Finally, the product was dried overnight in a vacuum oven at 40 °C.

### 2.3. Synthesis of $MAPbBr_3$ and $Mn/MAPbBr_3$ QD Dispersion

$MAPbBr_3$ QD dispersion was fabricated following the LARP method with modification (Figure 1) [13]. In detail, 0.225 mmol of MABr, 0.28 mmol of $PbBr_2$, and several molar ratios (from 0 to 4 mol%) of $MnBr_2$ vs. that of $PbBr_2$ were dissolved in 7 mL of 1-methyl-2-pyrrolidone as a polar solvent with 0.02 mmol of octylamine and 0.1 mmol of oleic acid,

resulting in obtaining a MAPbBr$_3$ or Mn/MAPbBr$_3$ QD precursor solution. A volume of 7 mL of MAPbBr$_3$ or Mn/MAPbBr$_3$ QD precursor solution was dropped into 25 mL of vigorously stirring chlorobenzene as a non-polar solvent and stirred for about 1 min. After stirring, as-synthesized MAPbBr$_3$ or Mn/MAPbBr$_3$ QD dispersions were centrifuged at 9000 rpm for 10 min and removed from the supernatants. Adding the same volume amount of ethyl acetate, it was centrifuged at 12,000 rpm for 10 min, and then the supernatant was discarded to remove excess ligands. After that, the precipitation was redispersed in toluene and centrifuged at 12,000 rpm for 5 min, and then a purified MAPbBr$_3$ or Mn/MAPbBr$_3$ QD dispersion was obtained.

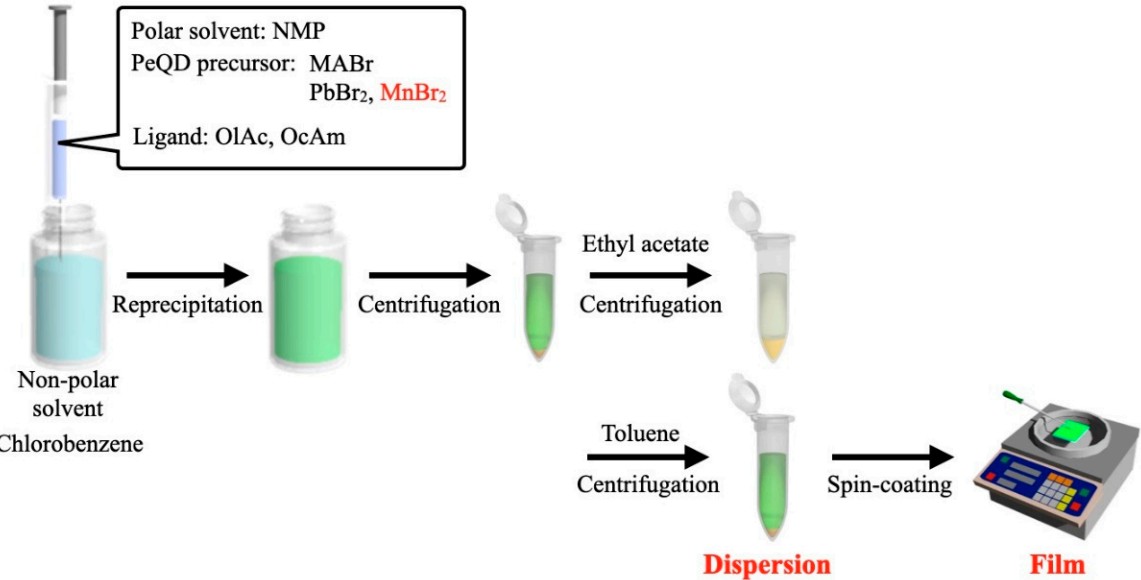

**Figure 1.** Preparing scheme of MAPbBr$_3$ and Mn/MAPbBr$_3$ QD dispersion and film by ligand-assisted reprecipitation and spin-coating methods.

### 2.4. Fabrication of MAPbBr$_3$ and Mn/MAPbBr$_3$ QD Thin Film

MAPbBr$_3$ and Mn/MAPbBr$_3$ QD dispersion was adjusted to a concentration of 10 mg mL$^{-1}$. By using purified MAPbBr$_3$ and Mn/MAPbBr$_3$ QD dispersion, MAPbBr$_3$ and Mn/MAPbBr$_3$ QD thin films were spin-coated at 2000 rpm for 30 s onto a glass substrate with a size of 2.5 × 1.5 cm$^2$ (Figure 1) [13].

### 2.5. Characterization

The morphologies of MAPbBr$_3$ and Mn/MAPbBr$_3$ QDs were observed by a JEOL JEM-2100F transmission electron microscope (TEM) (accelerating voltage of 200 kV). X-ray diffraction (XRD) patterns of MAPbBr$_3$ and Mn/MAPbBr$_3$ QDs were obtained from out-of-plane diffraction using membrane filters on a Rigaku Smart Lab using Cu Kα radiation at 45 kV and 200 mA. Visible absorption (UV) spectra of MAPbBr$_3$ and Mn/MAPbBr$_3$ QDs were obtained on a JASCO V-670 spectrophotometer (detecting wavelength range from 400 to 600 nm). Photoluminescence (PL) spectra and PLQYs were measured with JASCO FP-8600 to clarify the optical properties of MAPbBr$_3$ and Mn/MAPbBr$_3$ QDs. MAPbBr$_3$ and Mn/MAPbBr$_3$ QD thin films were stored in the temperature and humidity chamber (ESPEC CORP., bench-top type temperature & humidity chamber SH-222) under the condition of 25 degrees Celsius and 25% RH, and they were evaluated by PL spectra and PLQYs.

### 3. Results

#### 3.1. Morphologies of Prepared MAPbBr$_3$ and Mn/MAPbBr$_3$ QDs

To investigate the difference in the morphologies between MAPbBr$_3$ and Mn/MAPbBr$_3$ QDs, they were measured by TEM and XRD analysis. Figure 2 shows the TEM images of (a)

MAPbBr$_3$ or (b)–(e) Mn/MAPbBr$_3$ QDs with different molar ratios of the Mn component. From the TEM images, all prepared MAPbBr$_3$ or Mn/MAPbBr$_3$ QDs showed the mono-dispersity and dot-like shape. This indicated that there are no effects to Mn/MAPbBr$_3$ QD shape after containing an Mn component, resulting in successfully prepared Mn/MAPbBr$_3$ QDs. From XRD patterns, all prepared MAPbBr$_3$ and Mn/MAPbBr$_3$ QDs showed the diffraction peaks of (100), (110), (200), and (210) phase, which is attributed to the perovskite structure (Figure 3). Notably, focusing on the diffraction peaks, those of Mn/MAPbBr$_3$ with over 4 mol% of Mn component were drastically low angle-shifted, which implied that Mn was doped in the B site of the perovskite structure. An Mn$^{2+}$ ion could insert into the B site owing to its ion radius and tolerance factor which is the certain index of the PeQD structure stability. On the other hand, Mn/MAPbBr$_3$ with less than 4 mol% of Mn components (1–3 mol%) could not achieve their surface passivation on MAPbBr$_3$ because the diffraction peaks of the XRD patterns were not changed compared with that of MAPbBr$_3$ [24].

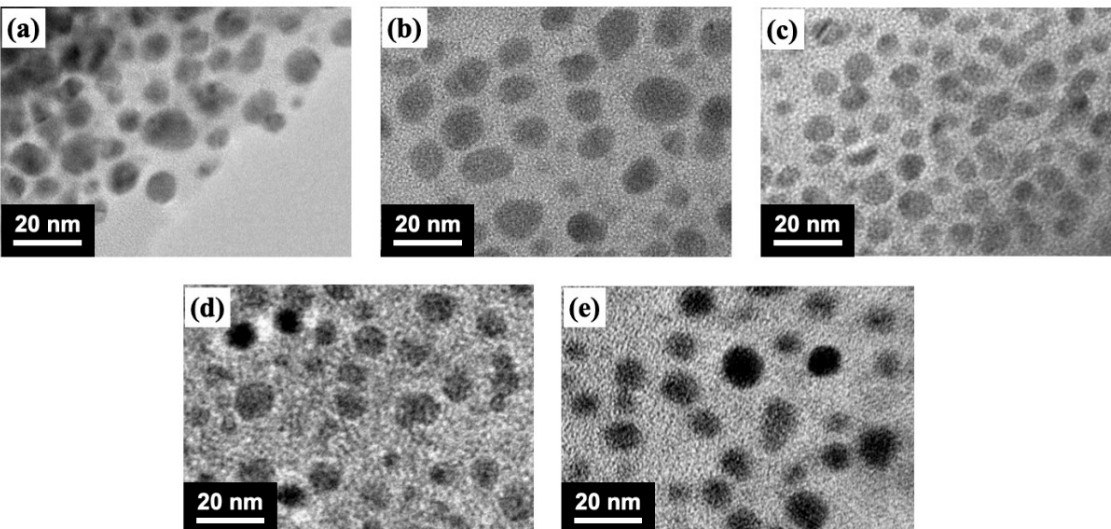

**Figure 2.** TEM images of (**a**) pure MAPbBr$_3$, and (**b**) 1 mol%, (**c**) 2 mol%, (**d**) 3 mol%, and (**e**) 4 mol% of Mn/MAPbBr$_3$ QDs.

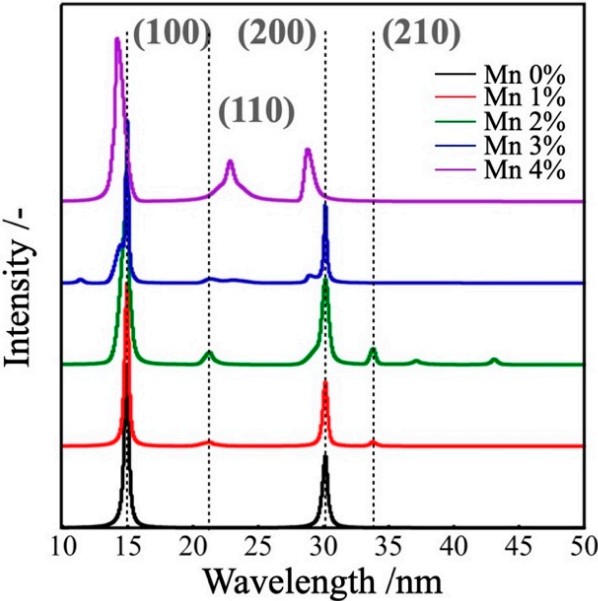

**Figure 3.** XRD patterns of MAPbBr$_3$ and Mn/MAPbBr$_3$ QDs.

### 3.2. Optical Properties of MAPbBr₃ and Mn/MAPbBr₃ QDs

To clarify the optical properties of MAPbBr$_3$ or Mn/MAPbBr$_3$ QD dispersion, their purified dispersion was evaluated. All prepared MAPbBr$_3$ and Mn/MAPbBr$_3$ QDs showed a bright green emission under 365 nm of ultraviolet irradiation (Table 1), and they showed specific absorbance attributed to PeQDs (Figure 4a) [25]. According to the PL spectra, with increasing the amount of Mn components from 1 to 4 mol% on Mn/MAPbBr$_3$ QDs, there was a gradually wider FWHM. In particular, the PL spectrum of 4 mol% Mn/MAPbBr$_3$ QDs was divided (Figure 4b). In the case of doping Mn into the B site on the perovskite structure, the crystal lattice is distorted because of the different ion radius between Pb (1.19 Å) and Mn (about 0.7–0.9 Å) at the B site. With increasing the ratio of distortion on the Mn/MAPbBr$_3$ structure, the B-X bonding state was generally changed, which is decided by the bandgap energy on PeQDs. As a result, the 4 mol% Mn/MAPbBr$_3$ QD emission wavelength was blue-shifted because the bandgap was wider, and the PL spectrum was divided, attributed to the crystal lattice distortion of Mn/MAPbBr$_3$ QDs [26]. In addition, MAPbBr$_3$ and Mn/MAPbBr$_3$ QD dispersion and film samples except for 4 mol% Mn/MAPbBr$_3$ QDs showed high PLQYs (Table S1). Dividing the bandgap caused the decrease of PLQYs because the different bandgaps generated by another B site worked as a trap site of electronic excitation. Moreover, focusing on the tolerance factor, Mn/MAPbBr$_3$ QD structure stability is expected to be lower with an increase in the presence of an Mn component at the B site on them owing to being an ion radius smaller [27]. Therefore, considering Mn/MAPbBr$_3$ QD structure stability and the B-X bonding state decreasing the PLQYs from 84 to 53% with increasing the Mn component is the correct result, which means the proof Mn presence into PeQDs in the case of the 4 mol% of Mn components.

**Table 1.** Images of MAPbBr₃ and Mn/MAPbBr₃ QD dispersion.

| | Mn 0% | Mn 1% | Mn 2% | Mn 3% | Mn 4% |
|---|---|---|---|---|---|
| Room light | | | | | |
| UV irradiation | | | | | |

### 3.3. Air Stability of MAPbBr₃ and Mn/MAPbBr₃ QDs

To prove whether the presence of the Mn leads to improved air stability of Mn/MAPbBr$_3$ QDs, their changing over time of the PLQY stability under the air condition was evaluated. MAPbBr$_3$ and Mn/MAPbBr$_3$ QD thin film were prepared by spin coating at 2000 rpm for 30 s to use 10 mg/mL of their purified dispersion and stored in the temperature and humidity chamber under the conditions of 25 degrees Celsius and 25% RH for 4 months. All the prepared films showed bright green emission under 365 nm ultraviolet irradiation (Table 2). Table S2 shows the images of the film changing over time. Surprisingly, Mn/MAPbBr$_3$ QDs have achieved higher air stability than that of the pure MAPbBr$_3$ QDs (Figure 5, Table S3). In detail, pure MAPbBr$_3$ thin film was quenched under 10% of PLQYs within 2 months (Figure 5 and S1). In contrast, Mn/MAPbBr$_3$ QD thin film which contained an Mn component of 1–4 mol% could maintain over 4 months. Considering the ionization tendency,

Mn is more easily oxidized compared with Pb. Therefore, with selectively passivating Mn on the surface of PeQDs, it is possible to rapidly form the passivation layer composed of oxidized Mn under an air atmosphere. As a result, an Mn/MAPbBr$_3$ QD thin film that endured the air atmosphere and humidity was successfully prepared.

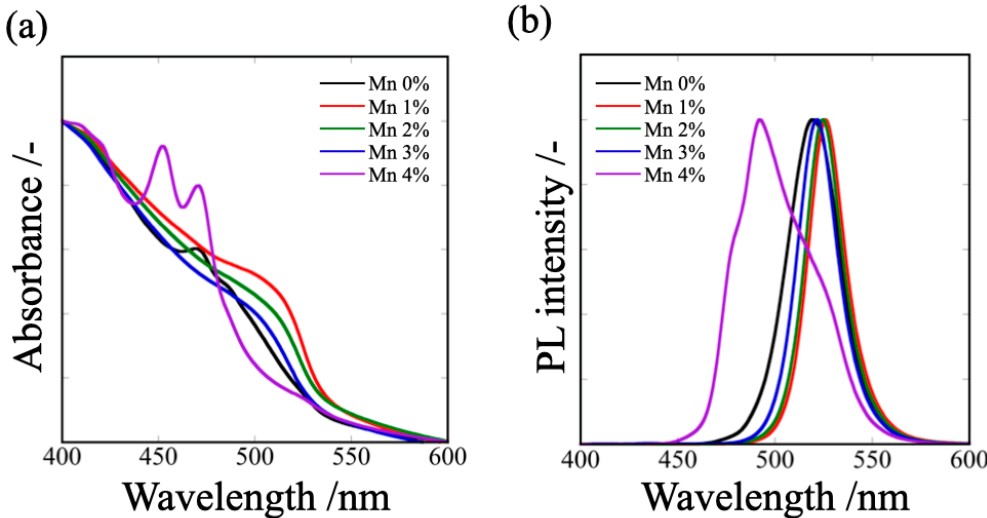

**Figure 4.** (**a**) Absorption and (**b**) PL spectra of MAPbBr$_3$ and Mn/MAPbBr$_3$ QD dispersion.

**Table 2.** Images of pure MAPbBr$_3$ and Mn/MAPbBr$_3$ QD films.

| | **Mn 0%** | **Mn 1%** | **Mn 2%** | **Mn 3%** | **Mn 4%** |
|---|---|---|---|---|---|
| UV irradiation | | | | | |

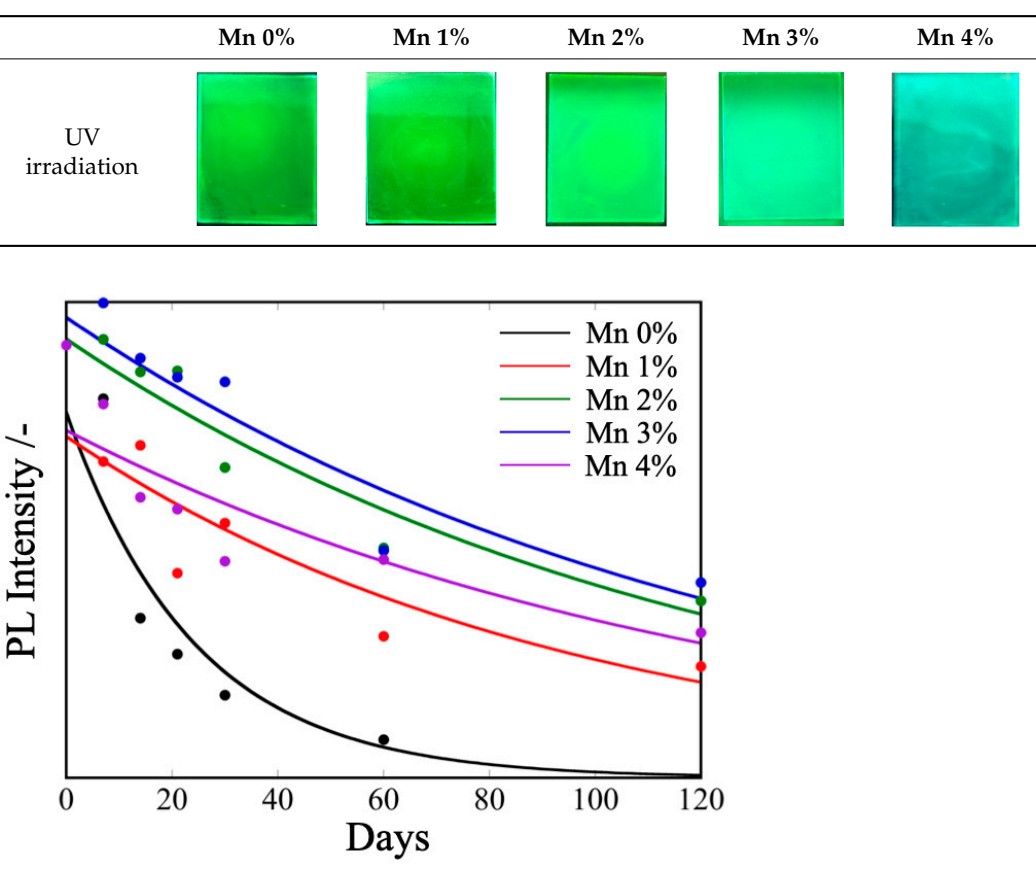

**Figure 5.** PL intensity (=air stability) of MAPbBr$_3$ and Mn/MAPbBr$_3$ QD thin film changing over time under the conditions of 25 degrees Celsius and 25% RH.

## 4. Conclusions

In summary, 1 to 4 mol% Mn/MAPbBr$_3$ QDs which are composed of passivated Mn were successfully prepared by LARP methods. They were uniform, dot-like shapes and constructed the perovskite structure. In addition, 1 to 4 mol% Mn/MAPbBr$_3$ QDs showed excellent optical properties. In particular, the changes over time of their PLQY stability under the air condition were significantly improved because of the presence of the oxidized Mn. These results suggested that the air stability of PeQDs was extremely improved with just only Mn passivation on MAPbBr$_3$ QDs. As conducted in this work, improving each PeQD stability promotes their longer life, which connects the development of optoelectrical applications.

**Supplementary Materials:** The following are available online at https://www.mdpi.com/article/10.3390/technologies10010010/s1, Figure S1: PL spectra of (a) pure MAPbBr$_3$, (b) 1 mol% Mn/MAPbBr$_3$, (c) 2 mol% Mn/MAPbBr$_3$, (d) 3 mol% Mn/MAPbBr$_3$, and (e) 4 mol% Mn/MAPbBr$_3$ QD film changing over time from 0 days to 4 months (120 days), Table S1: The optical properties of pure MAPbBr$_3$ or Mn/MAPbBr$_3$ QD dispersion; Table S2: The film state of pure MAPbBr$_3$ or Mn/MAPbBr$_3$ QDs changing over time; Table S3: The optical properties of MAPbBr$_3$ or Mn/MAPbBr$_3$ QD film.

**Author Contributions:** Conceptualization, A.M.; Methodology, R.S., K.U., S.A. and A.M.; Formal Analysis, R.S. and K.U.; Writing—Original Draft Preparation, R.S.; Writing—Review and Editing, R.S. and A.M.; Supervision, A.M.; Funding Acquisition, A.M. All authors have read and agreed to the published version of the manuscript.

**Funding:** This work was supported by the "Network Joint Research Center for Materials and Devices: Dynamic Alliance for Open Innovation Bridging Human, Environment and Materials 20211084 and 20214006".

**Institutional Review Board Statement:** Not applicable.

**Informed Consent Statement:** Not applicable.

**Data Availability Statement:** Not applicable.

**Conflicts of Interest:** The authors declare no conflict of interest.

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
