# Peer review of "Enhanced Air Stability of Perovskite Quantum Dots by Manganese Passivation"

_technologies, doi:10.3390/technologies10010010_

Round 1
Reviewer 1 Report
In general, the study by Sato et al. is interesting and complete. From a scientific point of view, I would like to see histograms made of TEM results PFig.2), because it looks that there is a change in size/size distribution. The authors claim that the size is 5-10 nm, which is true, but not enough.
Also, please define (made in the method section) the exact condition of temperature and humidity for storage tests. I know this is in the introduction, but anyway, something should be said about a box or whatever was used for storage.
There is a lot of (small but crucial) problems with language. See e.g. sentence in line 83. I believe, it should state "and white crystals were collected". Also, what exactly means thermal in line 68? A temperature? Stability over changing temperature? Looks like something is missing here. In lines 145-146 - why for film and dispersion properties, only the dispersion was checked (at the list, this is said here). The whole paragraph 3.2. is full of "their", and it is problematic to really find what every sentence is describing.
The paper should be definitely proofreader for clarity.
Reviewer 2 Report
This paper by Sato et al. reports the synthesis of stable perovskite quantum dots using Manganese passivation. The results and finding reported in this paper are of great interest to the readers of this journal. Therefore the manuscript can be accepted for publication after addressing the following concerns.
- The authors wrote "Mn passivated perovskite quantum dots", "Mn was changed", "Mn passivation" throughout the paper. However, the specific type of Mn should be determined and used in the paper. Are they Mn particles or Mn2+ or Mn4+ ions?
- Although authors mentioned "In detail, manganese (Mn) was chosen as a passivated layer.", the reason why they chose manganese was not presented in Introduction.
- I can see that Figure 1 provides many detailed information in the schematic illustration, but it looks too busy. The authors may attempt to revise.
- Figure 2 structure looks weird. The reason probably is due to the inclusion of 5 TEM images, leaving 1 empty spot. Can authors put Figure 3 in Figure 2 as Figure 2f?
- The following important and informative review articles on perovskite and lead chalcogenide quantum dots should be considered for discussion:
--- Shrestha et al. Near-Infrared Active Lead Chalcogenide Quantum Dots: Preparation, Post-Synthesis Ligand Exchange, and Applications in Solar Cells. Angewandte Chemie International Edition, 2019, 58, 5202-5224.
--- Lv et al. Improving the Stability of Metal Halide Perovskite Quantum Dots by Encapsulation. Advanced Materials, 2019, 31, 1900682.
Round 2
Reviewer 1 Report
I'm satisfied with the reviewed version of manuscript, I have no further comments.